# Respiratory Function Improvement and Lifespan Extension Following Immunotherapy with NP001 Support the Concept That Amyotrophic Lateral Sclerosis (ALS) Is an Immuno-Neurologic Disease

**DOI:** 10.3390/ijms26094349

**Published:** 2025-05-03

**Authors:** Rongzhen Zhang, Ari Azhir, Michael S. McGrath

**Affiliations:** 1Department of Medicine, University of California San Francisco, San Francisco, CA 94110, USA; 2Neuvivo, Inc., Palo Alto, CA 94303, USA

**Keywords:** ALS, inflammation, vital capacity (VC), body mass index (BMI), C-reactive protein (CRP), innate immunity, overall survival (OS), NP001

## Abstract

Amyotrophic lateral sclerosis (ALS) is a heterogeneous disease that involves progressive loss of voluntary muscle and ultimately, respiratory function, which is the primary cause of death in ALS patients. Respiratory vital capacity (VC) measurements are objective, reproducible, and directly related to survival. Respiratory function is known to be negatively affected in individuals with excess abdominal fat contributing to a chronic innate immune inflammatory state. To test whether ALS patients might have a body mass index (BMI) related VC response to the innate immune system regulator NP001, clinical results from two NP001 phase 2 trials were evaluated in an intent-to-treat manner, stratified by BMI measurements. Slowing of progressive VC loss and extension of overall survival (OS) occurred primarily in ALS patients who were overweight with a BMI ≥ 25 (70% of patients in the phase 2 trials). Innate immune dysfunction is a characteristic of ALS patients ≤ 65 years of age, and in this group both VC and OS changes in response to NP001 were most significant. This study represents a novel approach to ALS, wherein VC and OS were both significantly improved through immunologic, not neurologic modulation with NP001, a precursor to the dominant regulator of inflammation, taurine chloramine.

## 1. Introduction

The pathogenesis of amyotrophic lateral sclerosis (ALS) is complex, making targeted drug development difficult. Approximately 10–15% of patients have a family history of ALS, and this “familial” form of ALS has been associated with a broad spectrum of genetic abnormalities, the most common of which is the superoxide dismutase 1 (SOD1) gene. The finding of misfolded proteins such as TDP43 and pathogenic aggregation of those proteins have also failed to define a therapeutically validated approach to ALS [1]. Many studies have suggested a role for inflammation in ALS pathogenesis [2,3], but, to date, there are no disease-specific biomarkers identified nor anti-inflammatory approaches that have been successful in the treatment of ALS.

The testing of treatments for ALS relies on the use of the ALS functional rating scale-revised (ALSFRS-R), a 12-category, 4 points/category assessment scale that measures gross and fine muscle functions as well as bulbar and respiratory functions in a-48 point scale [4]. The ALSFRS-R scale is the primary tool for following disease progression, assessing efficacy of drugs and is predictive of survival. The other disease assessment process utilizes a single objective measure that is also associated with survival. This is the evaluation of respiratory function using forced vital capacity (FVC) or slow vital capacity (SVC) tests, both of which are objective and are predictive of survival. To date, even though respiratory failure is the principal cause of death, none of the approved ALS drugs have been shown to affect respiratory function.

Recent studies have identified disease-initiating activity as localized to neuromuscular junctions (NMJ) outside the central nervous system (CNS) [5]. This activity is mediated by blood-derived white blood cells participating in an innate immune response [6]. Multiple natural history population-based ALS incidence studies have confirmed this innate immune reaction as a risk factor for ALS development [7]. The dominant phenotype of this type of ALS patient is their age < 65 years old, body mass index (BMI) ≥ 25, and with blood showing evidence for innate immune activation.

NP001, a proprietary form of intravenously administered sodium chlorite, is under development for use in ALS and has as its principal mechanism of action the regulation of innate immune system activation. Chlorite, the active ingredient in NP001, is rapidly converted to intracellular taurine chloramine (TauCl) after activation by heme associated iron present in inflammatory cell molecules such as myeloperoxidase [8]. Intracellular TauCl is very long lived within macrophages, and it activates anti-inflammatory pathways driven by heme oxygenase 1 (HO-1) [9]. In vitro treatment of blood monocytes and macrophages causes a down regulation of proinflammatory factors and cell surface molecules such as HLA-DR and CD16. In a phase 1 dose ranging study NP001 administration down regulated both cell surface CD16 and HLA-DR after a single dose evaluated 24 h after administration [10]. The dose (2 mg/kg chlorite) chosen for the phase 2 studies was defined in part by the phase 1 study.

A summary of NP001’s activity in two underpowered, placebo-controlled, six-month phase 2 ALS trials was reported in early 2022 [3]. Post hoc analyses of the combined phase 2A and phase 2B trials defined a large subset of patients who responded clinically to NP001. The median plasma level of high-sensitivity C-reactive protein (hsCRP) in phase 2A was 1.13 mg/L, and patients above that value tended to respond better to NP001 [2]. The phase 2B recruited patients with hsCRP > 1.13 mg/L [3].

A recent study was published [11] that showed overall survival results from patients who participated in the two NP001 phase 2 trials. Phase 2A ended in 2012 and phase 2B ended in 2017, allowing a true long-term survival analysis. Actual dates of death or documentation of still living were obtained for 268 of the 274 patients who had at least one dose of drug with data collected and analyzed in a blinded manner. Predefined endpoints of an overall intent to treat (ITT) and for the age ≤ 65 years old subgroup, survival data were obtained for all ALS patients including those who received 1 mg/kg and 2 mg/kg NP001 and placebo. The ITT group showed an extension of survival of 4.8 months and the subgroup ≤ 65 years old participants 10.8 months with hazard ratios of 0.77 and 0.68, respectively, for NP001 treated with 2 mg/kg as compared to placebo. ALS patients who received 1 mg/kg or were > 65 years old showed no difference between treated and placebos.

The main reasons for re-evaluating all of the phase 2 data were the results from recent natural history studies identifying those patients at greatest risk for ALS. Having applied a hsCRP selection to participants in the phase 2B trial we were interested to see the relationship between the BMI and CRP values. Given the nonspecific nature of CRP and the many diseases associated with CRP elevation that are not ALS, we performed a separate analysis of these NP001 trials using BMI values using defined in the natural history studies as being associated with ALS. The NP001 clinical trial database including baseline hsCRP and overall survival (OS) values was analyzed using natural history dictated cutoff values of ≤ 65 years old with a BMI ≥ 25 [7].

## 2. Results

### 2.1. Differences of Demographics and Clinical Characteristics Between Low and High BMI ALS Patients in Phase 2A and 2B Modified Intend-to-Treat (mITT) Population

Table 1 shows the baseline demographics and ALS characteristics of the low and high BMI groups in the phase 2 mITT population, which includes all randomized subjects who received at least one dose of study infusion. There was no statistical difference between the ALS patients with baseline BMI < 25 and BMI ≥ 25 but a significantly lower number of El Escorial classification of possible ALS in the BMI < 25 group. A weak positive correlation was observed between baseline BMI and serum CRP levels (Pearson r = 0.25), and no correlation was found between BMI and vital capacity (VC) at baseline. However, baseline serum CRP levels from participants with baseline BMI ≥ 25 were significantly higher than those with BMI < 25 (Wilcoxon, *p* = 0.02) (Figure 1A). Figure 1B shows the baseline BMI distributions in the low and high BMI groups in the mITT population (Wilcoxon, *p* < 0.0001).

### 2.2. Effects of NP001 on VC and OS Are Significant and Beneficial in ALS Patients with BMI ≥ 25, but Not in Patients with BMI < 25 with Similar Demographics

In the phase 2A and 2B mITT population, change from baseline VC was evaluated in the high and low BMI groups. Figure 2A shows a significant slowing of VC loss over the six-month studies in the high BMI group treated with NP001. The NP001 treatment arm lost 23% less respiratory function than the placebo arm by the end of the study (NP001 = −9.3% vs. placebo = −12.1%) (Wilcoxon, *p* = 0.04). The change in VC over time did not differ by treatment status in participants with low BMI at baseline (Figure 2B, Wilcoxon, *p* = 0.88). Loss of VC was similar in both high and low BMI placebo groups.

In the high baseline BMI ALS patients, the median OS was 32.6 months and 29.3 months in the 2 mg/kg NP001 chlorite and placebo groups, respectively (log-rank, *p* = 0.04). Patients treated with NP001 had a better survival outcome than those on placebo (hazard ratio (HR) = 0.69, 95% CI: 0.49, 0.98), as shown in Figure 2C. No OS difference was observed between NP001 treatment and placebo arms in the BMI < 25 mITT population.

Note that no significant differences in baseline demographics or disease characteristics were observed by the two treatment arms in either the high BMI or the low BMI group. Table 2 shows patient demographics and disease characteristics of the phase 2A and 2B mITT population with BMI ≥ 25 at baseline.

### 2.3. OS and Percent of VC Change in mITT Population with High BMI and Age ≤ 65 Years Old at Baseline

Figure 3A shows the VC outcome during the clinical trials in ALS patients with BMI ≥ 25 and Age ≤ 65 years old at baseline. The NP001 treatment arm had 31% less respiratory function lost as compared to the placebo arm by the end of the study (NP001 = −9.4% vs. placebo = −13.6%) (Wilcoxon, *p* = 0.02).

In the ALS patients with BMI ≥ 25 and Age ≤ 65 years old, the median OS was 38.6 months and 33.1 months in the 2 mg/kg NP001 chlorite and placebo groups, respectively (log-rank, *p* = 0.008). Patients with ALS who were treated with NP001 had a 40% lower risk of death than those on placebo (HR = 0.60, 95% CI: 0.41, 0.87), as shown in Figure 3B.

No significant differences in baseline demographic and clinical characteristics were observed between the two treatment arms when analyses were restricted to the group aged ≤ 65 years old and BMI ≥ 25 at baseline (Table 3).

## 3. Discussion

The principle finding in the current study is related to a measurement, BMI, that provides two distinct reasons for loss of VC in patients with ALS. The first, that obesity itself can restrict pulmonary function. The second, that obesity is associated with inflammation in ALS pathogenesis. That VC is spared relative to placebo with NP001, BMI must represent the inflammation that would qualify as a target for NP001.

The measurement of body mass index is used routinely in clinical practice. A BMI greater than 25 is considered overweight and a BMI value of more than 30 defines “obese” individuals. In the current study, the majority of patients (>70%) fell into the high BMI subgroup. Even though demographically equivalent, there was a marked difference in clinical outcomes between ALS patients above and below a BMI of 25. In this study, this BMI cut point gave a population that was almost identical to a plasma CRP value of 1.13 mg/L to separate ALS patients for analysis [12].

In patients with ALS, loss of BMI pre-diagnosis defines a subset with the worst prognosis [13,14]. Historically, individuals with BMI values above 25 have a higher frequency of having low grade inflammation secondary to adipose production of inflammatory cytokines [15,16]. One of the properties of abdominal fat is that it is a source of chronic inflammatory factors that increase the pathogenic processes that are, in part, driven by inflammation that impacts pulmonary function [17]. CRP and CRP variants have also been implicated in contributing to inhibition of pulmonary function [18,19].

In previous studies, response to NP001 has been best in patients with above background levels of plasma CRP, specifically 1.13 mg/L [3]. In this inflammatory subset, the loss of vital capacity was slowed by over 50% and survival is presumed to have been extended. In the current study the inflammatory subset were those patients with a BMI ≥ 25. These patients showed both a slowing of VC loss and an extension of survival in those treated with NP001 compared with placebo. Importantly, the clinical benefits only showed in NP001 treated ALS patients with high BMI, but not low BMI. Considering that the demographics between low vs. high BMI are similar, inflammation must enable NP001 to react in a way that spares loss of VC and extends survival.

Recent natural history studies of risk factors for developing ALS have all found similar evidence for innate immune activation to be present predominantly in patients ≤ 65 years of age [7]. Blood measures of inflammation, such as elevation in granulocyte count, CRP or BMI in individuals who develop ALS also show that above the age of 65, ALS is not the only thing driving inflammation and disease outcomes. A natural history study of ALS performed in Sweden showed a clear relationship between age and CRP, with CRP increasing the longer the ALS patient had the disease, especially in those patients with innate immune dysfunction [20].

ALS patients in the subgroup of age ≤ 65 were shown to have a greater effect of NP001 on VC and OS [3,11]. In the study presented in this manuscript, ALS patients who had characteristics of innate immune dysfunction identified in the many natural history studies were evaluated for NP001 effects on VC and OS. ALS patients with a BMI ≥ 25 who were ≤ 65 years of age showed a 31% slowing in VC loss and an extension of OS of > 5 months. Slowing of VC loss and extending OS in ALS patients with a BMI ≥ 25 and age ≤ 65 years old again support NP001 as an innate immune system regulator.

BMI elevation gives the clinician an immediate indication that the ALS patient may have dysfunctional innate immunity as a component of their ALS disease. BMI elevation is associated with accumulation of abdominal fat, a source of proinflammatory factors that can contribute to ALS pathogenesis. The finding of plasma cytokine and lipopolysaccharide (LPS) levels to be abnormal [12,21,22] and likely contributing to microbial translocation (MT) confirms the role of innate immune activation in ALS pathogenesis, a process reversed in part by NP001.

## 4. Materials and Methods

### 4.1. Clinical Trials Overview

Two phase 2 trials were conducted by Neuraltus Pharmaceuticals, Inc. (Palo Alto, CA, USA) in ALS patients and were registered on ClinicalTrials.gov (phase 2A: NCT01281631, 28 February 2011, and phase 2B: NCT02794857, 29 August 2016). These were both placebo-controlled 6-month studies. Phase 2A was completed in 2012 [2], and phase 2B in 2017 [3]. No drug-related Serious Adverse Events (SAEs) occurred in either Phase 2A or 2B trials [2,3].

Both phase 2A and 2B studies were approved by the clinical site institutional ethics committees, and informed consent was obtained from all participants.

### 4.2. Description of ALS Phase 2A and 2B Trials and Participants

Details of these two six-month trials have been published [2,3]. For purposes of the current combined phase 2A and 2B studies, only patients treated with 2 mg/kg of NP001 chlorite or placebo were included. The dosage of NP001 used in both clinical trials is 2 mg/kg body weight as chlorite (equivalent to 2.682 mg/kg sodium chlorite).

In both studies, all patients were enrolled within 3 years of symptom onset. The additional criteria for all patients enrolled in the phase 2B study included a plasma high-sensitivity C-reactive protein (hsCRP) concentration of >1.13 mg/L at the pre-screening visit. The units expressed throughout the paper are hsCRP units, abbreviated as CRP. For each trial, patients were planned to receive a total of 20 infusions administered intravenously over 6 cycles during a 6-month study, with 4 weeks between the start of each cycle. Cycle 1 consisted of 30-min infusions over 5 consecutive days. Cycles 2, 3, 4, 5, and 6 each consisted of 3 consecutive 30-min daily infusions.

In the current study, we focused on patients with BMI ≥ 25 and treated with 2 mg/kg of NP001 chlorite or placebo in the modified intend-to-treat (mITT) population, which includes all randomized subjects who received at least one dose of study infusion. Two subjects without baseline BMI data available were excluded from the current study.

### 4.3. Analysis of Clinical Outcome Data

The current study focused on the percentage change in predicted vital capacity (VC) over the six-month studies. The phase 2A trial assessed the predicted VC in forced vital capacity (FVC), whereas phase 2B assessed the predicted VC as slow vital capacity (SVC). Since predicted VC values between FVC and SVC are comparable [23], the evaluation of NP001 effects on VC combined changes from both trials normalized to “% VC change from baseline [100 × (%predicted VC at study end − %predicted VC at baseline)/%predicted VC at baseline]”.

### 4.4. Phase 2A and 2B Survival Analyses

Survival data for ALS patients in the modified intention-to-treat (mITT) population were collected and recently reported by Forrest et al. [11]. These data were obtained from Neuvivo Inc. Overall survival (OS) was defined as the time in months from date of randomization to date of death due to any cause, or to last contact/known alive for patients lost to follow-up (censored) and were assessed through 30 September 2022.

### 4.5. Statistical Analyses

Statistical analysis was performed using JMP Pro 17 (SAS Institute, Cary, NC, USA) and packages “survival 3.7-0” [24] and “survminer 0.4.9” [25] in R 4.4.2 [26]. Data were summarized as counts and percentages for categorical data and using standard univariate descriptive statistics (number of participants, mean, standard deviation, median) for continuous/discrete data by treatment group. Preliminary analyses of categorical data were analyzed using Fisher’s exact test and Chi-square tests, and of continuous/discrete data were analyzed using Wilcoxon rank sum tests. Data that did not meet normality assumptions were log-transformed prior to analysis.

Overall survival differences by treatment/group were assessed using a log-rank test. Kaplan–Meier methods were used to estimate survival probabilities and curves. Cox-proportional hazards models were used to estimate the hazard ratio associated with dying for patients treated with NP001 compared with those given placebo. The assumptions of proportionality of hazards over time were assessed using the Schoenfeld test. Patients who were lost to follow-up or alive at the end of the follow-up period were considered censored in all survival analyses.

All statistical tests were two-sided and considered statistically significant for *p*-value < 0.05.

## 5. Conclusions

The data shown in this manuscript link aged ≤ 65, BMI ≥ 25, and change in VC with overall survival in a significant subset of ALS patients and confirm NP001 as a regulator of innate immune activation with persistent activity extending life significantly after the final dose of NP00. All of the data presented represents a post hoc analysis of data obtained from 2 placebo-controlled clinical trials. While the implications that immune dysfunction plays a role in ALS pathogenesis is suggested by the data presented, a follow-up study could better define the immune parameters that are dysfunctional in ALS.

### Future Perspectives

A major goal of this study was to determine whether a biological reason might exist to explain how an immunological intervention could have such a clinical effect on a major subset of ALS patients. The data presented was a mITT analysis of two groups of ALS patients that differed by one major physiologic variable, the BMI. Population based epidemiologic studies defined the greatest risk factor for the development of ALS as innate immune activation as documented in blood analyses. A major pathogenic process present at a higher rate in overweight and obese individuals is inflammation, historically linked to respiratory dysfunction. The data presented in this study are consistent with NP001 regulation affecting VC loss in high BMI but not in ALS patients with lower BMI whose VC progressive loss isn’t influenced by inflammation to the degree as in patients with a high BMI. The extension of survival in patients with high BMI linked to effects of NP001 on VC allows future studies to focus on a simple immunologic approach to slowing this devastating disease in the majority of patients in the US. Through immunologic innate immune manipulation, slow the loss of VC and extend life.

## Figures and Tables

**Figure 1 ijms-26-04349-f001:**
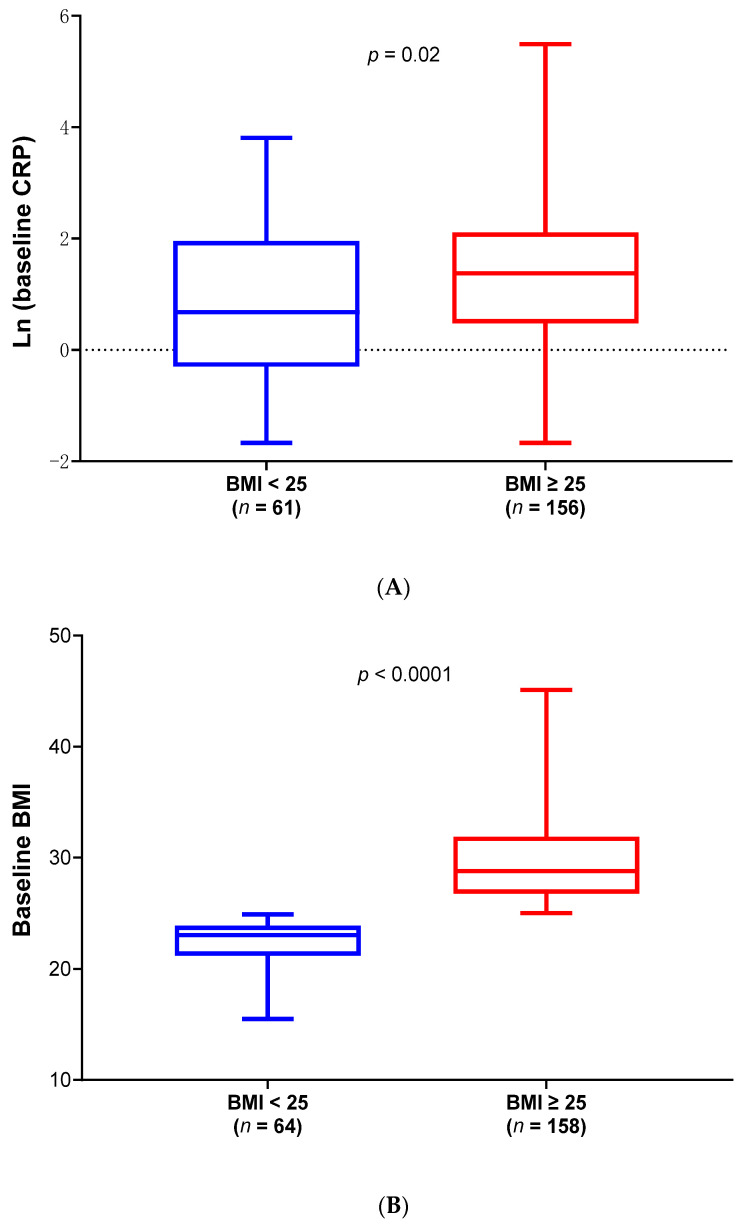
(**A**) High levels of baseline serum CRP observed in the high BMI group. Box and whisker plots depicting log-transformed CRP levels for the low BMI (BMI < 25, *n* = 61, in blue) and high BMI group (BMI ≥ 25, *n* = 156, in red). Results show that the median baseline CRP value was statistically significantly lower in the low BMI group compared with the high BMI group (Wilcoxon test, *p* = 0.02). Note: Median CRP of 1.600 mg/L and 2.596 mg/L, respectively, for BMI < 25 group compared to BMI ≥ 25 group. There was no baseline CRP available for 5 participants. (**B**) Box and whisker plots depicting the distribution of baseline BMI values for the low BMI group (baseline BMI < 25, *n* = 64, in blue) and the high BMI group (baseline BMI ≥ 25, *n* = 158, in red). Results show that the median baseline BMI value was statistically significantly lower in the low BMI group compared with the high BMI group (Wilcoxon test, *p* < 0.0001).

**Figure 2 ijms-26-04349-f002:**
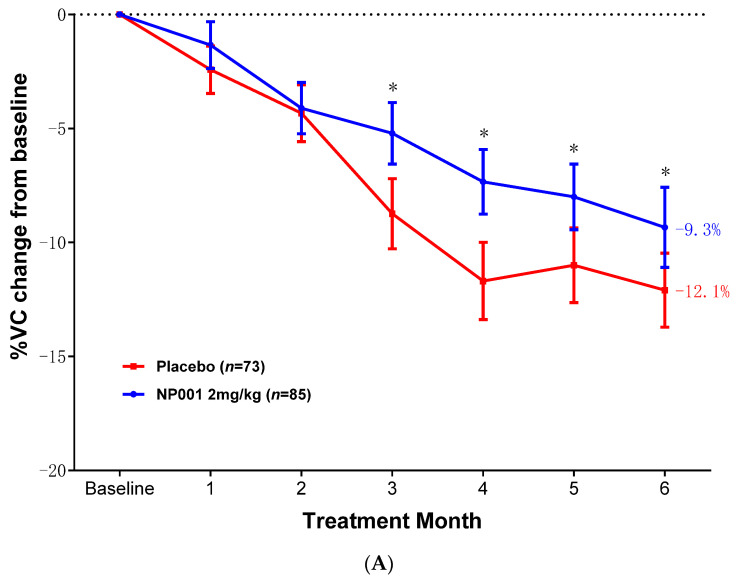
NP001 efficacy defined by survival analysis and %VC change in the mITT group with BMI ≥ 25 at baseline. (**A**) Change in % of VC from baseline over 6 months in participants with high BMI at baseline on NP001 compared with placebo. Percentage VC change from baseline for participants treated with NP001 (*n* = 85, blue) is compared with the placebo group (*n* = 73, red). Bars represent the mean of % VC change from baseline ± SEM. *, *p* < 0.05. Average %VC lost over the 6 months of study: NP001: −9.3% (−1.6% per month); placebo: −12.1% (−2.0% per month). The NP001 treatment arm lost 23% less respiratory function than the placebo arm by the end of the study (Wilcoxon test, *p* = 0.04). (**B**) Percentage change from baseline in VC over 6 months in participants on NP001 compared with placebo in those with BMI < 25 at baseline. Mean change from baseline in percent VC for participants treated with NP001 (*n* = 29, blue) compared with the placebo group (*n* = 35, red). Bars represent mean of % VC change from baseline ± SEM. There was no significant difference between NP001-treated patients and placebos by the end of the study (NP001 = −13.2% vs. placebo = −13.0%) (Wilcoxon test, *p* = 0.88). (**C**) Kaplan–Meier curve of survival probability for patients who received NP001 at a dose of 2 mg/kg compared with placebo in mITT high BMI population. The median survival over the entire follow-up duration was 32.6 months (95% CI: 28.3, 43.9) and 29.3 months (95% CI: 25.4, 40.0) in the 2 mg/kg NP001 (blue) and placebo (red) groups, respectively (log-rank, *p* = 0.04). Patients treated with NP001 had better survival than those on placebo: hazard ratio (HR) = 0.69 (95% CI: 0.49, 0.98).

**Figure 3 ijms-26-04349-f003:**
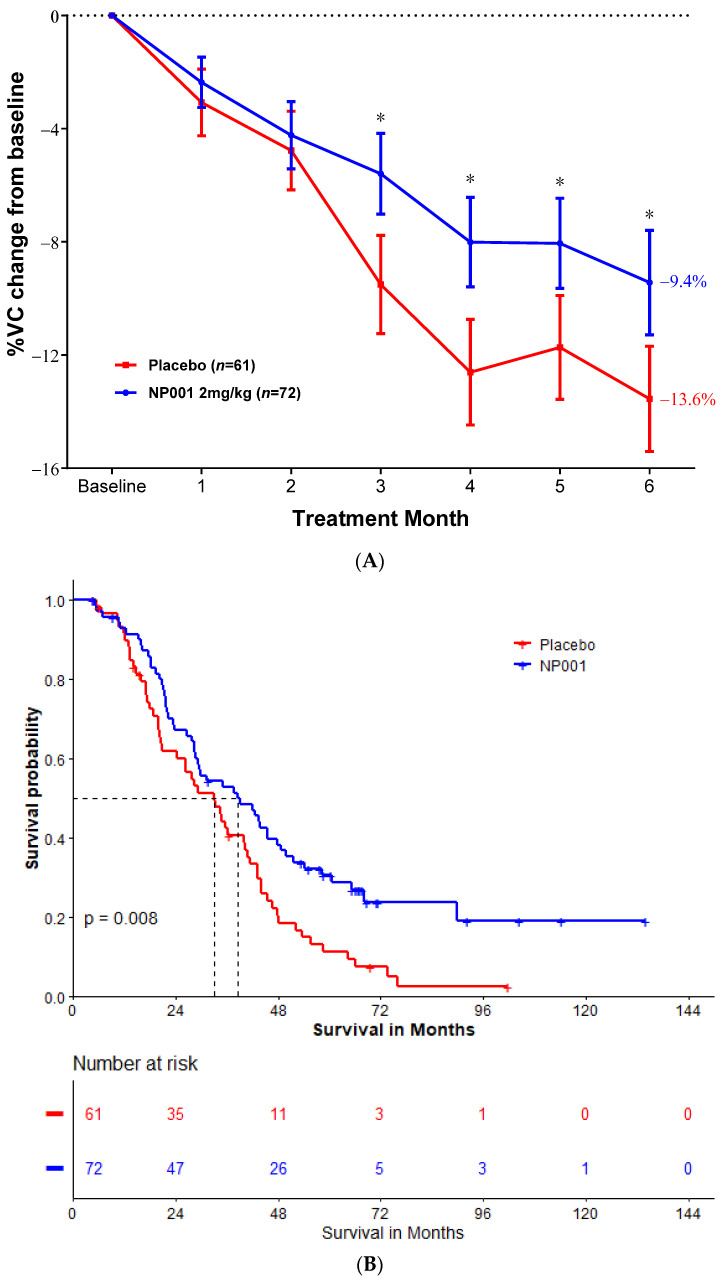
NP001 efficacy defined by survival analysis and %VC change in the mITT group with BMI ≥ 25 & age ≤ 65 years old at baseline. (**A**) Change in % of VC from baseline over 6 months in participants on NP001 compared with placebo with high BMI & age ≤ 65 at baseline. Percentage VC change from baseline for participants treated with NP001 (*n* = 72, blue) is compared with the placebo group (*n* = 61, red). Bars represent the mean of % VC change from baseline ± SEM. *, *p* < 0.05. Average %VC lost over the 6 months of study: NP001: −9.4% (−1.6% per month); placebo: −13.6% (−2.3% per month). The NP001 treatment arm lost 31% less respiratory function than the placebo arm by the end of the study (Wilcoxon test, *p* = 0.02). (**B**) Kaplan–Meier curve of survival probability for patients who received NP001 at a dose of 2 mg/kg compared with placebo in high BMI & age ≤ 65 mITT population. The median survival over the entire follow-up duration was 38.6 months (95% CI: 28.8, 49.9) and 33.1 months (95% CI: 24.4, 41.6) in the 2 mg/kg NP001 (blue) and placebo (red) groups, respectively (log-rank, *p* = 0.008). Patients treated with NP001 survived longer than those on placebo: hazard ratio (HR) = 0.60 (95% CI: 0.41, 0.87).

**Table 1 ijms-26-04349-t001:** Baseline demographics and characteristics of mITT population ^1^ with BMI < 25 and BMI ≥ 25 at baseline.

	BMI < 25	BMI ≥ 25	Overall
Characteristics	(*n* = 64)	(*n* = 158)	*(n* = 222)
Sex, *n* (%)			
Female	22 (34.4%)	50 (31.6%)	72 (32.4%)
Male	42 (65.6%)	108 (68.4%)	150 (67.6%)
Age at baseline, year	56.3 ± 11.4	55.7 ± 10.3	55.9 ± 10.6
Site of ALS onset, *n* (%)			
Bulbar	14 (21.9%)	18 (11.4%)	32 (14.4%)
Limb	50 (78.1%)	140 (88.6%)	190 (86.6%)
El Escorial criteria for ALS, *n* (%)			
Definite	27 (42.2%)	68 (43.0%)	95 (42.8%)
Possible	1 (1.6%) *	16 (10.1%)	17 (7.7%)
Probable	34 (53.1%)	62 (39.2%)	96 (43.2%)
Probable Laboratory Supported	2 (3.1%)	12 (7.6%)	14 (6.3%)
Concurrent riluzole use, *n* (%)			
Yes	43 (67.2%)	106 (67.1%)	149 (67.1%)
No	21 (32.8%)	52 (32.9%)	73 (32.9%)
ALSFRS-R score at baseline, mean ± SD	36.7 ± 5.7	37.8 ± 5.2	37.5 ± 5.4
Vital capacity at baseline, mean ± SD	96.6 ± 20.5	93.1 ± 19.9	94.1 ± 20.1
Months since ALS symptom onset ^2^, mean ± SD	17.65 ± 8.59	18.12 ± 8.10	17.98 ± 8.23

Abbreviation: *n*, number of participants. SD, standard deviation. ^1^ Modified intend-to-treat (mITT) population, all randomized subjects who received at least one dose of study infusion. ^2^ Months from ALS symptom onset to baseline. * *p* < 0.05.

**Table 2 ijms-26-04349-t002:** Baseline demographics and characteristics of mITT population ^1^ with BMI ≥ 25 at baseline.

	NP001 2 mg/kg	Placebo	Overall
Characteristics	(*n* = 85)	(*n* = 73)	*(n* = 158)
Sex, *n* (%)			
Female	28 (32.9%)	22 (30.1%)	50 (31.6%)
Male	57 (67.1%)	51 (69.9%)	108 (68.4%)
Age at baseline, year	55.5 ± 10.5	55.9 ± 10.1	55.7 ± 10.3
Site of ALS onset, *n* (%)			
Bulbar	8 (9.4%)	10 (13.7%)	18 (11.4%)
Limb	77 (90.6%)	63 (86.3%)	140 (88.6%)
El Escorial criteria for ALS, *n* (%)			
Definite	35 (41.2%)	33 (45.2%)	68 (43.0%)
Possible	9 (10.6%)	7 (9.6%)	16 (10.1%)
Probable	35 (41.2%)	27 (37.0%)	62 (39.2%)
Probable Laboratory Supported	6 (7.1%)	6 (8.2%)	12 (7.6%)
Concurrent riluzole use, *n* (%)			
Yes	59 (69.4%)	47 (64.4%)	106 (67.1%)
No	26 (30.6%)	26 (35.6%)	52 (32.9%)
ALSFRS-R score at baseline, mean ± SD	38.4 ± 4.9	37.2 ± 5.5	37.8 ± 5.2
Vital capacity at baseline, mean ± SD	95.6 ± 20.0	90.1 ± 19.5	93.1 ± 19.9
Months since ALS symptom onset ^2^, mean ± SD	18.03 ± 7.90	18.22 ± 8.39	18.12 ± 8.10
BMI, mean ± SD	29.7 ± 3.7	29.9 ± 4.2	29.8 ± 3.9

Abbreviation: *n*, number of participants. SD, standard deviation. ^1^ Modified intend-to-treat (mITT) population, all randomized subjects who received at least one dose of study infusion. ^2^ Months from ALS symptom onset to baseline.

**Table 3 ijms-26-04349-t003:** Baseline demographics and characteristics of mITT population ^1^ with BMI ≥ 25 and age ≤ 65 years old at baseline.

	NP001 2 mg/kg	Placebo	Overall
Characteristics	(*n* = 72)	(*n* = 61)	*(n* = 133)
Sex, *n* (%)			
Female	23 (31.9%)	17 (27.9%)	40 (30.1%)
Male	49 (68.1%)	44 (72.1%)	93 (69.9%)
Age at baseline, year	52.8 ± 9.0	53.1 ± 8.5	53.0 ± 8.7
Site of ALS onset, *n* (%)			
Bulbar	6 (8.3%)	9 (14.8%)	15 (11.3%)
Limb	66 (91.7%)	52 (85.2%)	118 (88.7%)
El Escorial criteria for ALS, *n* (%)			
Definite	31 (43.1%)	30 (49.2%)	61 (45.9%)
Possible	7 (9.7%)	7 (11.5%)	14 (10.5%)
Probable	29 (40.3%)	20 (32.8%)	49 (36.8%)
Probable Laboratory Supported	5 (6.9%)	4 (6.6%)	9 (6.8%)
Concurrent riluzole use, *n* (%)			
Yes	48 (66.7%)	38 (62.3%)	86 (64.7%)
No	24 (33.3%)	23 (37.7%)	47 (35.3%)
ALSFRS-R score at baseline, mean ± SD	38.6 ± 4.8	37.0 ± 5.5	37.8 ± 5.2
Vital capacity at baseline, mean ± SD	96.7 ± 20.5	91.4 ± 18.7	94.3 ± 19.8
Months since ALS symptom onset ^2^, mean ± SD	18.73 ± 8.08	18.00 ± 8.36	18.39 ± 8.19
BMI, mean ± SD	29.7 ± 3.7	30.1 ± 4.3	29.8 ± 4.0

Abbreviation: *n*, number of participants. SD, standard deviation. ^1^ Modified intend-to-treat (mITT) population, all randomized subjects who received at least one dose of study infusion. ^2^ Months from ALS symptom onset to baseline.

## Data Availability

The data are available through Neuvivo, Inc. upon request.

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
