# Peer review of "Respiratory Function Improvement and Lifespan Extension Following Immunotherapy with NP001 Support the Concept That Amyotrophic Lateral Sclerosis (ALS) Is an Immuno-Neurologic Disease"

_ijms, 2025, doi:10.3390/ijms26094349_

Round 1
Reviewer 1 Report
Comments and Suggestions for Authors
The manuscript idea is excellent and interesting; however, few points should be considered to make it clearer:
1- The title needs to be more precise and avoid using abbreviations
2- The introduction needs to cover the NPOO1 mechanism in details. The authors should state in a separate paragraph the aim of the study
3- The results miss the signs of significance in all tables and graphs
Was any statistical correlation performed?! I guess it is better to perform correlation between BMI and CPR, %VC and BMI
Author Response
Response to reviewer 1’s comments
The manuscript idea is excellent and interesting; however, few points should be considered to make it clearer:
- The title needs to be more precise and avoid using abbreviations
Thanks to the reviewer for the great suggestion. The title has been modified to “Respiratory Function Improvement and Lifespan Extension Following Immunotherapy with NP001 Supports the Concept that Amyotrophic Lateral Sclerosis (ALS) is an Immuno-Neurologic Disease”.
- The introduction needs to cover the NPOO1 mechanism in details. The authors should state in a separate paragraph the aim of the study
A significant section of the introduction in the 4th paragraph has been included to cover the MOA of NP001 as suggested.
The aim of the study has been described in the last paragraph in Introduction.
- The results miss the signs of significance in all tables and graphs
As the reviewer suggested, the signs of significance, * and text, table footnote/figure legends, have been modified as needed to reflect the signs of significance.
- Was any statistical correlation performed?! I guess it is better to perform correlation between BMI and CPR, %VC and BMI
Yes, we have performed statistical correlations. The results have been included in the first paragraph in Results section 2.1 as follows: “A weak positive correlation was observed between baseline BMI and serum CRP levels (Pearson r = 0.25), and no correlation was found between BMI and VC at baseline.”
Reviewer 2 Report
Comments and Suggestions for Authors
Dear authors,
The article is interesting; however, major revision is required for publication in the journal.
Major comments:
-Please, add more detail about amyotrophic lateral sclerosis (ALS) pathogenesis in the Introduction section.
-Please add a graphical abstract to give to the reader an overview of the model.
-Please, highlight the future perspectives of the study.
- Please check all abbreviations in the manuscript.
Comments on the Quality of English LanguageThe English language could be improved.
Author Response
The article is interesting; however, major revision is required for publication in the journal.
Major comments:
-Please, add more detail about amyotrophic lateral sclerosis (ALS) pathogenesis in the Introduction section.
As suggested, we have expanded the introduction to include pertinent information related to ALS pathogenesis and how the current study approaches this process in a novel manner, as seen in the 1st & 2nd paragraphs.
-Please add a graphical abstract to give to the reader an overview of the model.
We attempted to construct a graphical abstract without success. The concept of innate immune activation and regulation with NP001 contributing to slowing of VC loss in ALS in high but not low BMI individuals linked to survival outcomes wasn’t easily displayable. Hopefully the revised title will help the reader understand the purpose and outcome of the study.
-Please, highlight the future perspectives of the study.
We have added a section at the end with a future perspectives focus.
- Please check all abbreviations in the manuscript.
Done. Thanks for the reviewer’s suggestion.
Round 2
Reviewer 2 Report
Comments and Suggestions for Authors
The authors improved the manuscript